# A Program for the Comprehensive Cognitive Training of Excess Weight (TRAINEP): The Study Protocol for A Randomized, Controlled Trial

**DOI:** 10.3390/ijerph19148447

**Published:** 2022-07-11

**Authors:** Lucía Solier-López, Raquel González-González, Alfonso Caracuel, Naomi Kakoschke, Natalia Lawrence, Raquel Vilar-López

**Affiliations:** 1Mind, Brain and Behavior Research Center (CIMCYC), University of Granada, 18070 Granada, Spain; luciasl@ugr.es (L.S.-L.); raquelgonzalez@ugr.es (R.G.-G.); rvilar@ugr.es (R.V.-L.); 2Nutrition and Health Program (Health & Biosecurity) at CSIRO, Adelaide, SA 5000, Australia; naomi.kakoschke@csiro.au; 3Department of Psychology, University of Exeter, Exeter EX4 4PY, UK; natalia.lawrence@exeter.ac.uk

**Keywords:** excess weight, obesity, cognitive training, approach–avoidance bias, inhibitory control, implementation intentions, episodic future thinking

## Abstract

Background: The available treatments for people with excess weight have shown small effects. Cognitive training has shown promising results, but most of the research focused on normal-weight university students and reported immediate results after a single training session. This parallel group, randomized, controlled trial aims to study the efficacy of a program for the comprehensive cognitive treatment of excess weight. Methods and Analysis: Participants will be 150 people with excess weight recruited through social media, who will be randomized into three groups: cognitive intervention, sham cognitive intervention, and treatment as usual. All assessment and intervention sessions will be online in groups of 5–6 participants. The three groups will attend a motivational interviewing session, and they will receive individualized diet and physical exercise guidelines throughout the program. The cognitive training will consist of four weekly sessions of approximately 60–90 min, each based on approach–avoidance bias training, inhibitory control training, implementation of intentions, and episodic future thinking, respectively. The main outcome measure will be a change in Body Mass Index (kg/m^2^). Secondary outcomes include changes in cognitive measures, eating and physical exercise behaviors, and anthropometric measures. Assessments will be conducted up to 6 months after the end of the program. In addition, data on the use of the health system will be collected to analyze the cost-effectiveness and the cost-utility of training. Linear mixed models will be used for statistical analysis. Findings of this study will expand the available evidence on cognitive interventions to reduce excess weight.

## 1. Introduction

Obesity prevalence has increased in the last few years until reaching pandemic levels [1]. Obesity and overweight are risk factors for respiratory viral infections, such as COVID-19, cardiovascular disorders, and immune and endocrine system deficits [2], as well as depressive symptoms [3], body image dissatisfaction, and stress [4]. In general, physical and psychological problems derived from obesity trigger a diminished quality of life [5]. Furthermore, the increase in obesity and its associated health problems is very costly to healthcare services [6]. The most frequent treatments for obesity and overweight, such as dietitian counselling, physical exercise, pharmacology, or surgery, are not effective enough in the long term [1,7]. Thus, it is important to pair them with psychological interventions, being cognitive–behavioral therapies (CBT), the cornerstone of excess weight treatment; CBT works on increasing healthy eating and healthy activity, and it also relies on cognitive–behavioral strategies (such as stimulus control or cognitive restructuring) to reinforce weight management behaviors. Nevertheless, one meta-analysis indicates that adding CBT increases weight loss of active treatments by only 1.7 kg [8]. In short, the global increase in overweight and obesity along with the limited efficacy of existing treatments reveals the necessity to find new efficient and effective treatments that target the mechanisms underlying excess weight, such as cognitive training strategies.

In recent years, neuroscience has been trying to determine which brain and cognitive characteristics are related to excess weight. It has been found that people with excess weight have alterations in ‘bottom-up’ (impulsive) and ‘top-down’ (executive) systems associated with difficulties in self-regulation [9]. The dysregulation between the two systems is characterized by high impulsivity and poor inhibition, making it difficult to self-regulate eating behaviors and leading to overeating and weight gain [10,11,12,13]. Cognitive training strategies aimed at changing these imbalanced systems have been reviewed for their individual effectiveness [14], and the results have shown their potential for instilling healthy habits and weight loss [15]. Some of these cognitive strategies for unhealthy eating and obesity include approach–avoidance bias modification [16], response inhibition training [17], implementation of intentions [18], and episodic future thinking [19].

The approach–avoidance training (AAT) addresses the approach–avoidance bias that is a fast automatic approach response to appetitive stimuli and a fast response of avoidance to aversive stimuli [20]. People with excess weight show this approach–avoidance bias to unhealthy and healthy food images, respectively [21]. Accordingly, training tasks to modify this bias consist of approaching images of healthy food on a screen and moving away images of unhealthy food [21]. Findings have shown the efficacy of the ATT in excess weight participants, but its effect for the long term is unknown. A reduction in the approach to unhealthy food images with stable effect several days after the ATT has been found [22]. Additionally, ATT produced an improvement in a food choice task immediately after the training [23]. Diary ATT during 7 days through the application Tilt Task has also shown a reduction of the bias to approach unhealthy food, plus an increase in eating healthy food [16]. Similarly, six AAT sessions showed improvements in self-regulation, food consumption, and quality of life in 6-month follow-ups, but these improvements were not maintained 12 months later [24]. Recent meta-analysis with normal and excess weight samples concluded that AAT is effective to reduce the approach–avoidance bias and that it is perceived as effective by participants [25]. Despite these promising results, it is necessary to conduct clinical trials to determine the effects of AAT to improve real eating behavior (beyond reductions in approach–avoidance bias) [25]. 

Regarding the inhibitory control training (ICT), it is aimed at inhibiting the automatic response to unhealthy food intake shown by people with excess weight [12]. In the Go/No-Go training paradigm, healthy food images are linked with the go response (press a key) and unhealthy food images with the No-go response (not to press the key). One session of ICT was able to diminish the consumption of sweets and tasty food assessed after the training session [26] as well as to improve food choice [23]. Four ICT sessions have been effective to reduce daily intake and weight and devaluation of unhealthy food [27]. A 2-week ICT achieved a reduction of unhealthy food consumption and an increase of healthy food that was maintained one week after the training [28]. Furthermore, ICT for sweets consumption produced weight loss in people with sweet preferences in food [29]. Although ICT has demonstrated good results, effect sizes are small and therefore combination with other strategies is recommended [30]. 

The implementation of intentions training (IIT) aims to turn a previous decision (based on a motivation to change) into an action [31] by establishing plans with a format “If… then…” (e.g., “If I am bored along the afternoon, then I will eat an apple”) [14]. Establishing sufficiently well-specified plans is the mechanism by which the IIT works [32], by increasing congruence of behaviors with the objective of losing weight [33]. The IIT has shown a higher effect to increase healthy food than to decrease unhealthy food [34,35]. This strategy has been effective for people with a BMI > 27 to lose weight in a 10-week program [18]. Conclusions of a systematic review considered the II an effective option to diminish energy intake and to lose weight in people with excess weight [14]. 

In the episodic future thinking training (EFTT), people establish clear and specific goals in the future that facilitate actions in the present that are congruent with that future. In short, health goals are linked with future events defined with all possible details—who, what, how, where–(for example, “Next week I will go to the concert with my friend, and we will stay near the stage to enjoy our favorite music group. I will feel happy and proud of myself for achieving my goal of running twice a week, and I will enjoy the concert”) [19]. EFTT has been shown to improve delay discounting [36,37,38], diminish energy intake [36], reduce fat energy intake, and increment proteins [19] in people and also in families [39] with excess weight. Nevertheless, other studies did not find a positive effect [16,40]. Thus, further research is guaranteed.

To conclude, the results of the efficacy of cognitive training for excess weight are promising, albeit with small effect sizes [4,14,25]. Most research focused on normal-weight university students, finding immediate results after one single training session. Thus, clinical trials to study the effect of combined cognitive training strategies and long-term follow-ups in people with overweight and obesity are still needed [14,25,30,41]. The protocol of this trial was designed to overcome these limitations. The overall aim is to determine the effectiveness of a comprehensive program that incorporates four cognitive techniques including two targeting the impulsive system (approach–avoidance training and inhibitory control training) and two targeting the reflexive system (implementation of intentions and episodic future thinking). This program will be compared to its sham version and a treatment as usual for reducing excess weight. The specific aims are: 

To determine the effectiveness of a comprehensive cognitive training to reduce Body Mass Index (BMI) in the short, medium, and long-term (post-treatment and 3- and 6-month follow-ups). 

To examine the effectiveness of the comprehensive program to change eating behaviors, physical activity, and anthropometric measures.

To determine the predictive or mediating role of the variables related to cognition, affect, stress and non-homeostatic food intake, motivation, and personality to explain the intervention outcomes.

To conduct an economic evaluation analysis of cost-effectiveness and the cost-utility of the training and to analyze the budgetary impact for the Spanish National Health System.

We hypothesize that the comprehensive cognitive program will be more effective for the treatment of overweight and obesity than the sham and the treatment as usual. Thus, the cognitive program will produce a greater reduction in BMI. In addition, the cognitive program will improve other anthropometric measures (waist circumference, waist-to-hip and waist-to-height ratios), lifestyle behaviors (eating behavior and physical activity), as well as cognitive processes (approach–avoidance bias, inhibition control, food liking and valuation, and delay of gratification). Further, the program will show its efficiency in the economic balance of cost-effectiveness and cost-utility.

## 2. Materials and Methods

### 2.1. Experimental Design, Setting, and Dates

The Program for the comprehensive cognitive training of excess weight TRAINEP (by its acronym in Spanish: Programa para el TRAtamiento cognitivo INtegral del Exceso de Peso) is a double-blinded, randomized, controlled trial with parallel groups. The study is based at the Mind, Brain, and Behavior Research Center (CIMCYC) at the University of Granada (Granada, Spain). However, all interaction with participants during the trial is online through the services contracted by the University of Granada on the platforms GoogleMeet (videoconference during the evaluation and intervention sessions) and LimeSurvey (submission of the answers to the evaluation instruments administered during the assessment sessions). In addition, as part of the intervention, the participants will receive messages on their mobile phones. The study was registered at www.clinicalTrials.gov number NCT05158075. Data collection started in March 2021 and will be completed by June 2023. 

### 2.2. Sample Size Calculation

The calculation of the sample size has been made from Objective 1, which focuses on detecting a clinically significant difference in the BMI, the study’s primary outcome measure. Based on previous work that explored weight change in follow-ups after the application of cognitive training in people with overweight and obesity [16,39,42], we have selected the smaller sample size (Cohen’s *d* = 0.46). With this data, considering the statistical method (models of repeated measures inter and intra subjects), the number of groups (g = 3), and assuming an alpha level of 0.05 and a power of 0.80, the resulting sample size calculated with G-Power v3.1.9.7 (Heinrich Heine University Düsseldorf, Düsseldorf, Germany) was 105 participants. To date, no study has conducted a program based on comprehensive cognitive trainings that have been proposed as effective for the treatment of obesity, with a 6-month follow-up period. Thus, we opted for a conservative estimate increasing the number of participants to 150. Then, the study could suffer an attrition rate of 30% from the initial session to the follow-up at 6 months and the statistical power would still be guaranteed. Recent studies that explored cognitive training to lose weight have shown similar attrition rates (26% in Stice et al. [25]).

### 2.3. Recruitment, Participants, Eligibility Criteria and Blinding 

Participant recruitment will be carried out through social media and the website of the project (trainep.ugr.es). The trial will include people between 18 and 55 years old with a good command of the Spanish language, a BMI between 25 and 39.9 kg/m^2^, and with at least two electronic devices available (one tablet, computer or smartphone to attend the online meetings, and one smartphone with a gyroscope to do the training with the applications). 

Exclusion criteria include: (i) traumatic, digestive, metabolic, or systemic disorders that affect the central nervous, autonomic or endocrine systems, (ii) cardiovascular or any other disorders that prevent physical exercise; (iii) psychopathological disorders or presence of severe symptoms in the Depression Anxiety and Stress Scale-21 (DASS-21); (iv) eating disorders or presence of DSM-5 criteria in the Questionnaire on Eating and Weight Patterns-5 (QEWP-5); (v) pharmacological or any other kind of treatment for losing weight at present; (vi) candidates for bariatric surgery; (vii) food conditions that could interfere with the stimuli in the program (allergies, sprue, vegetarianism, veganism); (viii) current pregnancy or breastfeeding (or expected pregnancy in the following six months); (ix) weight loss > 5% during the 3 months previous to the program; (x) current use of psychiatric or any other medication that affects weight or food intake (fluoxetine, olanzapine, etc.); (xi) frequent use of alcohol (>3 days a week) or use of other drugs that affect food intake. All exclusion criteria will be evaluated by self-report questionnaires (see the Outcome Measures section). Nevertheless, individualized clinical interviews could be done when research psychologists suspect any exclusion condition.

A randomization by minimization process will be performed through the software Minimizer^®^ to avoid imbalances between the groups in age, sex, and BMI [43]. Thus, participants (N = 150) will be randomly allocated to one of three groups: Experimental (verum cognitive training; *n* = 50), Sham (sham cognitive training; *n* = 50), and TAU (treatment as usual; *n* = 50). 

The psychologist that conducts the assessments (screening, evaluation sessions, and follow-ups) will be blinded to the group allocation during the whole project. Moreover, since all assessments will be computerized and online, they are not subject to the biases of the evaluator. Further, all participants will be blind to their condition. Additionally, the people who perform the statistical analyses will be blind to the condition of the participants through the coding of the interventions.

### 2.4. Outcome Measures

The measures have been selected whenever possible based on ADOPT (Accumulating Data to Optimally Predict Obesity Treatment) [44] to generate evidence about the efficacy for obesity interventions. These measures are marked with an asterisk (*) below. The assessments are conducted across five different visits throughout the study: pretreatment, mid-treatment, post-treatment, and 3-month and 6-month follow-ups. A full schedule of assessments is provided in Figure 1. All measures taken with Visual Analogue Scales have been named by adding the abbreviation VAS.

#### 2.4.1. Main Outcome Measure

BMI*: Height and weight will be obtained from a pharmacy digital scale to calculate the Body Mass Index (kg/m^2^). The BMI will be used to determine the results of the whole program (the four intervention sessions), and it will be registered weekly from session 1 to post-treatment assessment and at follow-ups (3- and 6-months post-treatment). In addition, the week before the pre-treatment assessment session, it will be recorded daily and then averaged to obtain the baseline weight.

#### 2.4.2. Secondary Outcomes

##### Changes in Cognitive Processes

Approach–avoidance bias: The approach–avoidance bias will be assessed before and after the 1-week training with the Tilt Task. The assessment will be done for all participants through the sham version of the Tilt Task application. Participants will use their smartphone to zoom in or out of the unhealthy and the healthy food images that appear on the screen one at a time, depending on the presentation format of the image (horizontal or vertical). Approach and avoidance trials will account for 50% of each type of image (healthy vs. unhealthy). Note that this percentage is different to that of the active training application in which 90% of healthy food images appear in the format to be approached. The median reaction time of the approach (pull) and avoidance (push) trials will be subtracted separately for healthy and unhealthy foods, with the positive and the negative scores representing approximation and avoidance bias, respectively [see Kakoschke et al. [16] for a full description].Inhibition control: This process will be assessed before and after the 1-week training with the Food Trainer. The assessment will be done for all participants through the sham version of the Tilt Task application. A total of 50% of the healthy and the unhealthy foods are paired with the Go and 50% with the No-Go signal. Inhibitory control will be determined by the effect of learning the association between the No-Go signal and unhealthy foods. For this, we will calculate an error learning rate by subtracting the error rate on No-Go trials of unhealthy foods from the error rate on No-Go trials of non-food images. The learning rate will be higher the lower the error rate with unhealthy foods relative to non-food images [17].Food valuation. Food Liking Task [27]: Participants will be presented with the images of the foods trained on the Food Trainer and asked to imagine that they are in their mouth to rate how much they like the taste with a visual analogue scale anchored at the extremes “not at all” and “very much”. Participants will move a cursor along the scale using a mouse and press the mouse button to confirm their rating, and the score between 0 and 100 will be recorded. The score will be the mean value of each type of food (healthy/unhealthy).Food Choice Task [16]: In this task, 16 pictures of different healthy and unhealthy snack foods previously trained on the Tilt Task will be presented. Participants will select eight of these foods. They will have 15 s to make their choices. The scores will be the number of healthy foods selected as a proportion of the total number selected.Delay of gratification*: The Now or Later questionnaire [45] will be used to measure sensitivity to immediate rewards versus higher-value delayed rewards at different time intervals at pre-intervention. At post-intervention, for the experimental group each item of the questionnaire will be applied immediately after reading the trained episodic future thinking cue, attending to the temporary interval of the item [46]. For the active control group, each item of the questionnaire will be applied immediately after reading the recent episodic thinking cue trained, attending to the temporary interval of the item. The control group will respond to the Now or Later questionnaire without any modification. 

##### Changes in Lifestyle (Behaviors and Adherence to Healthy Eating and Physical Activity)

Self-reported diary. Participants will complete a diary for 7 days to report:
The frequency and quantity of snacks consumed;The frequency and duration of physical exercise;The desire to snack and do physical exercise;The perceived level of achievement of the intentions (only in the experimental/sham groups).Self-reported 72 h food intake [47]: Units and quantities on the amounts of foods and drinks that the participants eat and drink along these hours will be registered in a pencil and paper format and then transformed into the number of calories for carbohydrates, sugars and fat, as well as total energy intake.Mediterranean Diet Adherence Screener (MEDAS) [47]: A score on the 14 items that evaluate healthy nutritional habits (e.g., fresh fruits and vegetables, olive oil, etc.). International Questionnaire about Physical Activity (IPAQ) [48]: A score on this questionnaire is from 7 questions about physical activity during the last week as well as time walking and sitting down. Number of daily steps (pedometer): A pedometer (an app on the smartphone or smartwatch) will be used to register the daily number of steps through the whole program.Healthy eating with a Visual Analog Scale (VAS), with the question: ‘During the last week, how healthy do you think your diet has been?’ Healthy eating is understood as that according to the information received in the program: a low amount of ultra-processed food, a low amount of added sugar or alcohol; and a high amount of whole grain products, fruits, and vegetables; etc. (0 = not healthy at all; 100 = 100% healthy).Physical exercise habits with a VAS, with the question: ‘During the last week, what has been your level of activity and physical exercise?’ (0 = no activity at all; 100 = maximum level of activity that I would be able to do).

##### Changes in Anthropometric Measures

Body composition measures are obtained from the pharmacy digital scale used to measure weight and height (see main outcome measure):Fat percentage;Water percentage;Muscle mass percentage.

Furthermore, participants are instructed on how to obtain the next measures by themselves with a tutorial video:Waist circumference* (WC): Participants should stand with heels close together and trunk erect, and put the tape measure around the waist, just above the navel, to measure the waist circumference in centimeters.Waist-to-hip ratio* (WHR): Participants should stand with heels close together and trunk erect. To measure waist circumference, participants should put the tape measure around the waist, just above the navel. To measure hip circumference, participants should place the tape measure at the maximum gluteal prominence. The hip-to-waist ratio is determined by dividing the waist circumference (in cm) by the hip circumference (in cm). Waist-to-height ratio (WHtR): Participants should stand with heels close together and trunk erect, and put the tape measure around the waist, just above the navel, to measure the waist circumference in centimeters. The height in centimeters will be measured with a digital weight. The waist to hip ratio is determined by dividing the waist circumference by the height.

#### 2.4.3. Predictive and Mediating/Moderating Variables

##### Affect, Stress, and Non-Homeostatic Eating

State affect. Positive and Negative Affect Schedule* (PANAS) [49]: The PANAS contains 10 items to evaluate a negative affect experienced in the last week and another 10 items to evaluate a positive affect.Perceived stress. Perceived Stress Scale* (PSS) [50]. A score on this scale with 10 items evaluates stress feelings experienced during the last month. Emotional eating. Coping subscale of the Palatable Eating Motives Scale* (PEMS-coping) [51]: A score on four items that evaluate the intentionality for eating palatable foods to face negative emotions.Trait food craving. The Food Craving Questionnaire Trait-reduced* (FCQ-T-r) [52]: A score on this questionnaire evaluates the tendency to experience food cravings.Reward-related eating. Reward-Based Eating Scale* (RED) [53]: A score on this scale with 13 items evaluate worries about foods, losing intake control, and the absence of satiety. Non-homeostatic eating. Dutch Eating Behavior Questionnaire (DEBQ) [54]: A score on this questionnaire assesses restrictive eating behaviors related to external cues and emotional states.

##### Self-Reported Executive Functioning

g.Behavior Rating Inventory of Executive Function-Adults Version* (BRIEF-A) [55]: A score on this questionnaire assesses cognitive dysfunction in behavioral and emotional actions in everyday life. It consists of 75 items divided into 9 subscales: inhibition, changes, self-monitoring, initiation, working memory, planning, monitoring task, material organization, and emotional control.h.Consideration of Future Consequences scale- short food version (CFC) [56]: A score on 12 items that assess a temporal perspective related to health and nutrition, with items oriented to the immediate ("I only act to satisfy immediate concerns, figuring the future will take care of itself") and the future (“I consider how things might be in the future, and try to influence those things with my day-to-day behavior”).

##### Motivation

i.Behavioral intention to diet. Dieting Intentions Scale* (DIS) [57]: A score on this 7-item scale concerns the intention to follow a diet.j.Behavioral intention to exercise. Physical Exercise Intentions Scale (PEIS): A score on this 7-item scale about the willingness to do physical exercise. This scale is an adaptation from the Dieting Intentions Scale.k.Self-efficacy. Weight Lose Self-efficacy scale* (WLSE) [58]: A score is obtained on 12 items that assess the self-confidence to eat healthy food, do physical exercise, and lose weight.l.Hedonic response to food ‘liking’. Food Liking* VAS: Participants will answer the question: ‘How much do you like ultra-processed food (i.e., high calorie foods, high in sugars and fats, like pizza, hamburgers, chocolate cake, chips, etc.)?’ Responses have a 0 to 100 scale, where 0 = I don’t like it and 100 = I absolutely like it.m.Hedonic response to motivation for food. Food Wanting* VAS: Participants will answer the question: ‘How much do you want ultra-processed food (i.e., high calorie foods, high in sugars and fats, like pizza, hamburgers, chocolate cake, chips, etc.)?’ Responses have a 0 to 100 scale, where 0 = I don’t want it at all and 100 = I absolutely want it.n.Hunger. Hunger* VAS through the question: ‘How hungry do you feel?’ with a 0 to 100 scale, where 0 = not hungry at all and 100 = absolutely hungry.o.Motivation to change. The Stages of Change Readiness and Treatment Eagerness Scale (SOCRATES 00; Vieira da Silva et al., 2020): Scores on this questionnaire relate to motivations to change adapted to excess weight. It has 18 items that score readiness to change in people with abusive food use.

##### Personality

p.Big Five factors. Mini International Personality Item Pool* (mini-IPIP) [59]: A score on 20 items that evaluate the big five factors of Personality (extraversion, agreeableness, conscientiousness, neuroticism, and openness).q.Impulsivity. Impulsive Behavior Scale (UPPS-P [60]. This scale evaluates five personality factors that can trigger impulsive behaviors: negative and positive urgency, lack of premeditation, lack of perseverance, and sensation seeking. All 20 items are scored on a scale from 1 (I agree strongly) to 4 (I disagree strongly).r.Inhibition and activation systems. Sensitivity to punishment and reward. Punishment Sensitivity and Reward Sensitivity Questionnaire (PSRSQ) [61]: Score on this questionnaire with 48 dichotomous response items (Yes/No). The instrument has two subscales of 24 items each: The Punishment Sensitivity subscale, related to the inhibition behavioral system, and the Reward Sensitivity subscale, related to the activation behavioral system of Gray’s theory.

##### Adherence to Diet and Physical Exercise

s.Adherence to diet with a VAS, with the question: During the last week, what has been your degree of compliance with the eating guidelines given by the nutritionist of the program? (0 = I did not comply with the diet at all; 100 = I absolutely complied with the diet).t.Adherence to physical exercise with a VAS, with the question: During the last week, what has been your degree of compliance with the physical exercise guidelines given by the physical trainer of the program? (0 = I did not comply with the exercise recommendations at all; 100 = I absolutely complied with the exercise recommendations).

##### Clinical Variables

u.Participants’ will answer having suffered weight stigma (Yes/No) [62], rate their motivation to intervention (1 to 5), and state the number of previous interventions to lose weight.

#### 2.4.4. Screening and Descriptive Measures

Sociodemographic (age, education, sex, socioeconomic variables), BMI, and clinical variables to consider exclusion criteria (central nervous system, autonomic, or endocrine disorders; cardiovascular disorders; treatment for losing weight; bariatric surgery; conditions that interfere with the stimulus of the program, weight loss > 5% on the 3 months previous to the program; medication that affects weight; use of alcohol or other drugs);Depression Anxiety and Stress Scale-21 (DASS-21) [63]: Scores on the 3 subscales that evaluate anxiety, depression, and stress;Questionnaire on Eating and Weight Patterns-5 (QEWP-5) [64]: Score on the items of the questionnaire that is adapted to DSM-5 criteria. It will be used to exclude people with binge eating problems and bulimia.

#### 2.4.5. Measures to Calculate Cost Effectiveness, Cost Utility, and Budget Impact Analysis

SF-36 Quality of Life Questionnaire: Total score on this questionnaire will be used to estimate the quality of life in terms of utility. The utility will be estimated based on the tariff validated for Spain [65];Quality-adjusted life years (QALYs). The QALYs is the most used measure in economic evaluation. It is a measure composed of years of life and profits (collected from the SF-36) that reflect the quality of life of the population under study. This measure will be used in the cost-utility analysis of the intervention;Body Mass Index (kg/m^2^). This measure will be used to calculate the cost effectiveness of the neurocognitive training intervention;Cost of health resources that the participants spend (visits to primary care, emergencies, hospital admissions, and medicines consumed);Cost of the time spent by the personnel in charge of the neurocognitive training sessions. A professional with the category of Clinical Psychology for each session of cognitive training with 5 participants, with 1 h 30 min per session for 5 weeks. The cost per hour will be collected according to the salary of the personnel of health centers and institutions of the Public Health Service.

### 2.5. Procedure and Interventions

All sessions will be conducted online in manageable groups of 5–6 participants. Eligible participants will attend an information meeting in which they will receive written and verbal information about the project, and they will be asked for their informed consent. Then, participants will be randomly assigned to groups. The three groups of the study will complete all of the evaluations, as well as the follow-ups (see below and Figure 1). Further, all groups will receive the same usual interventions, comprising motivational interviewing (for increasing adherence and to provide information about nutrition and physical activity to match the participants’ knowledge), an individualized diet and physical exercise guidelines, and nutrition and physical exercise sessions to resolve doubts. What will differentiate the groups will be, therefore, the cognitive treatment (verum vs. sham vs. no cognitive treatment). All procedures will be conducted by psychologists, except the nutrition and physical exercise plans, which will be conducted by a Ph.D. nutritionist and a master’s degree physical exercise professional, respectively. 

The experimental and the sham intervention programs consist of four weekly sessions of approximately 60–90 min organized in two types of interventions (impulsive and executive) whose order will be counterbalanced. One task will be introduced and trained in each of the four sessions and participants will practice the task at home for 7 days. 

#### 2.5.1. Interventions Targeting the Impulsive System 

Approach–avoidance training with the Tilt Task app [16]. In this app participants are instructed to zoom in or out of a food image, according to the image format (vertical or horizontal), by tilting their smartphone or tablet toward or away from them. In the experimental condition, 90% of healthy food images appear in the format to be approached (for example, vertical), while only 10% of unhealthy images appear in that format (to ensure concentration on the task). In the sham condition, 50% of healthy food images and 50% of unhealthy food images appear in the format to be approached (which is the same as in the assessment version).Inhibition training with the Food Trainer task application [17]. This application applies a Go/No-Go training. The images are presented and immediately after a green (go signal) or red circle (no-go signal) appears around them. Participants are instructed to touch images with a green circle as quickly as possible but to not press on the red circle images. A total of 50% of images are food (25% healthy, 25% unhealthy) and 50% are non-foods. From 15 food categories (default, alcohol, biscuits, bread, cakes, cheese, chocolate, crisps/chips, fast food, fizzy drinks, ice-cream, meat, pastries, pizza, and sweets), participants can choose which three unhealthy food categories they would like to train to resist. In the experimental condition, pictures of healthy and unhealthy foods are always paired with the Go and the No-Go signal, respectively, while in the sham condition both types of foods are paired 50% of the time with the Go and the No-Go signal. Non-food images are paired 50% of the time with the Go and the No-Go signal in both versions of the application. 

These two applications should be practiced at least once a day during the training week of each task. Participants receive one reminder to practice the task on their mobile phones every day (participants in the sham condition receive this reminder at 1 p.m. and participants in the experimental condition receive this reminder at the time of the day they refer as most problematic with food). 

#### 2.5.2. Interventions Targeting the Reflexive System

c.Implementation of Intentions technique [18] establishing one intention related to food and one related to exercise. Participants in the experimental condition should write an action plan detailing when, where, and how they will implement each one, the inconveniences that they may experience, and how they will overcome them. Furthermore, motivational cues (‘why I eat?’) will be considered (e.g., emotions like sadness or pleasure, social motives, politeness/conformity with others, expectations, and so on). Implementation of intentions related to food will be written in the format: ‘If… then…’. This format allows for the consideration of the health goals of each participant (e.g., ‘If I come home from work hungry in the evening, then I will eat an apple’). The intention related to food is focussed on reducing unhealthy snacking habits whenever possible, but another intention can be selected whenever snacking is not a relevant objective to the participant. Participants in the sham condition receive a session focussed on setting intentions, where participants choose 10 items of healthy foods they could eat whenever they want to eat unhealthy food, and they also order a list of physical activities that they could do to lose weight, according to their personal preferences. For one week, participants must read their action plans to promote the implementation of their intentions (experimental group) or read their food and exercise lists (sham condition) at least twice a day, when they receive the reminder on their phone (at 7:00 a.m. and 5:00 p.m.).d.Episodic Future Thinking [19] for the experimental condition consists of deciding health-related goals and linking them to anticipated positive future personal events in three time periods (next month, 2–3 months, and 4–6 months). Participants value from 0 to 5 their goals (“how important and accessible is the goal?”) and events (“how positive and vivid can I imagine the event?”). Thus, the best valued goal and event of each period is selected. After that, participants must detail the contextual, temporal, and emotional cues of each event selected. Then, goals and future events are matched to generate episodic future thinking cues. Participants in the sham condition conduct a session of Recent Episodic Thinking that consists of writing habits and positive recent past events in three time periods (1 day ago, 2 days ago, 1 week ago), both unrelated to food and physical activity. Participants value their habits from 0 to 5 (‘how important is the habit for me?’) and events (‘how positive and vivid can I remember the event?’). Thus, the highest valued habit and event of each period are selected. After that, participants must detail the contextual, temporal, and emotional cues of each event selected. Then, habits and events are matched to generate recent episodic thinking cues. The psychologist tells the participants that the objective of the task is to recall the positive emotions associated with these cues. Finally, the visualization of the cues is practiced. For one week, all participants must visualize their three cues at least twice a day, when they receive the reminder on their phone (at 7:00 a.m. and 5:00 p.m.), to promote decision-making according to long-term health goals in the experimental group. Participants must respond to the second daily reminder, evaluating from 0 to 10 how much they have kept their cues that day to ensure adherence.

#### 2.5.3. Compliance and Adherence

In both experimental and sham groups, different strategies are implemented to improve adherence to intervention protocols throughout the program: all groups attend a motivational interview session to promote the therapeutic alliance and adherence to the program. All of the intervention sessions are opened with an explanatory exposition to ensure that participants correctly understand the task. In addition, the correct execution of each task is assured in each session: The Tilt Task application has an initial trial with non-food images (e.g., stationary) to practice the task before proceeding with the real trial. Furthermore, in both Tilt Task and Food Trainer sessions, the researcher asks participants to show their smartphone screen to ensure that movement is detected correctly. As well, if any participant has problems perceiving colors, the Food Trainer application allows him/her to substitute this variable for another (continuous vs. discontinuous line). Regarding reflexive training sessions, the researcher asks the participants to send the worksheets after the group session ends. Thus, the researcher can check the correct execution and suggest improvements to each participant individually. Regarding homework compliance, daily reminders are sent by WhatsApp (thus, the researcher can check that they are received correctly). Daily training at home is recorded in different ways: Each practice with Tilt Task and Food Trainer applications is automatically recorded in the online database; referring to Implementation of Intentions, participants should complete and send us a daily self-registration. Lastly, during the Episodic Future Thinking training week, participants should answer and send us some questions on WhatsApp every day. In addition, session participants answer a short debriefing at the beginning of each intervention about their practice over the previous week to monitor the daily task execution. Besides, every month after the end of the treatment, participants will be contacted by email and mobile message to explore possible difficulties in following healthy nutritional and exercise habits and to ask them about their physical activity and achievement of health goals. The purpose of this contact is, fundamentally, therapeutic adherence. 

### 2.6. Statistical Methods

The applied inferential statistics will be conducted in accordance with the characteristics of the data obtained (distribution of the data, qualitative/quantitative nature of the data, etc.) and with the hypotheses proposed in the study. Models of repeated measures (pretreatment, post-treatment, and 3-month and 6-month follow-ups) two-way ANOVAs will be performed for Objectives 1, 2, and 3. Effect sizes for between-group pairwise comparisons will be calculated using the equation by Morris [66] to estimate the magnitude of the difference between three groups at different points in time. Within-group effect sizes will be calculated using Cohen’s *d* [67], and based on his rule of thumb, clinical significance of change will be reflected by medium within-group effect sizes (*d* ≥ 5). Mediation and/or moderation analysis will be carried out to study the role of BMI and variables related to cognition, affect, stress and non-homeostatic food intake, motivation, and personality on the program outcome measures. In addition, possible differences in the program outcomes will be assessed based on sex and weight classification (overweight and obesity—type I and type II). 

For Objective 4, the incremental cost effectiveness ratios will be calculated by dividing the difference in total cost by the difference in measures of effectiveness (BMI and QALYs) for the three groups. All analyses will be carried out according to the methodological recommendations for Spain [68]. The perspective of the health system will be used. To assess the robustness of the results, various deterministic sensitivity analyses (AS) and a probabilistic analysis will be performed, using nonparametric bootstrapping methods of 1000 iterations [28] that will allow representation of the acceptability curve. The AIP will estimate the variation that the introduction of cognitive training would imply for the budget of the public health system, based on the estimated costs of the alternatives compared and the estimated number of patients expected to benefit from the program. Likewise, univariate and multivariate AS will be performed to analyze different budgetary scenarios based on a possible variation in the cost variables and the number of beneficiary patients. 

All the analyses described will be carried out following an intention to treat strategy (ITT) and analysis per protocol (PP). Corrections will always be considered to control for multiple comparisons. All analysis will be conducted using the IBM SPSS v28 Statistics software [69] and the significance level will be set at *p* < 0.05. 

## 3. Expected Results

The comprehensive program based on experimental knowledge about the functioning and the modification of the impulsive and the executive systems will be effective in overweight and obesity treatment. Cognitive verum training—compared to sham training and treatment as usual—will reduce BMI and approach–avoidance bias, and it will improve inhibitory control and healthy eating and exercise behavior in the medium to long term. Finally, the program will show its effectiveness in the economic balance of cost-effectiveness and cost-utility.

## 4. Conclusions 

The aim of this double-blind, randomized, controlled trial with parallel groups is to examine the efficacy of an integrative cognitive treatment in people with excess weight versus a sham or no cognitive treatment. The challenges of the study are to examine the long-term outcomes (6 months) of four different combined trainings in an overweight population. The international ADOPT consensus has been applied to select outcome measures in order to generate evidence easily comparable with the efficacy of other obesity interventions.

## Figures and Tables

**Figure 1 ijerph-19-08447-f001:**
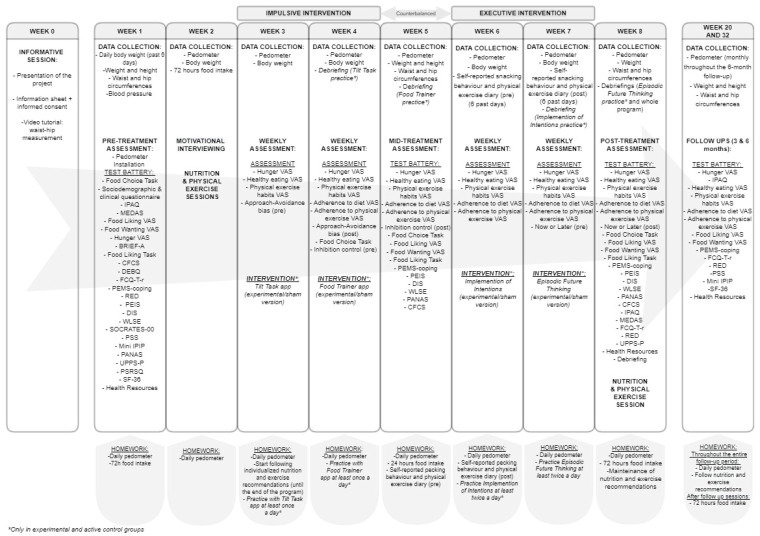
Full schedule of assessments and interventions.

## Data Availability

Study data will be made available from the corresponding author upon reasonable request.

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
