# Peer review of "A Program for the Comprehensive Cognitive Training of Excess Weight (TRAINEP): The Study Protocol for A Randomized, Controlled Trial"

_ijerph, 2022, doi:10.3390/ijerph19148447_

Round 1

Reviewer 1 Report

The article describes an ambitious randomized controlled trial aims to demonstrate the efficacy of a complex program on weight loss using  comprehensive cognitive training. The expected number of participants seems sufficient for the three proposed groups of intervention, with both the recruitment and the intervention sessions occurring online. The final follow-up should be done at 6 months after intervention.

The proposed assessment and intervention sessions seem comprehensive and a bit lengthy, with much data available for collection and with potential to generate interesting results, but this might be a concern for patient retention on the long run. I suggest a larger number of included participants from the beginning.

There should be introduced a method of evaluating the real readings of the pedometers as some passive activities might generate false movement. Where smartwatches are available, maybe it would be useful to use the body composition analysis already available on the smartwatch and also collect this data in relation to bodyweight and BMI.

Some corrections of the English language should be done. For example, "pedometer" is the correct term (not podometer) that should be used in the table describing the trial interventions and proposed data collection; line 242 "0 = not healthy at all", or line 310 "How much do you want ultra-processed food?" Please thoroughly revise the questions and questionnaires that will be presented to the participants before starting the recruitment process.

Author Response

Point 1: The article describes an ambitious randomized controlled trial aims to demonstrate the efficacy of a complex program on weight loss using comprehensive cognitive training. The expected number of participants seems sufficient for the three proposed groups of intervention, with both the recruitment and the intervention sessions occurring online. The final follow-up should be done at 6 months after intervention.

The proposed assessment and intervention sessions seem comprehensive and a bit lengthy, with much data available for collection and with potential to generate interesting results, but this might be a concern for patient retention on the long run. I suggest a larger number of included participants from the beginning.

Response 1: Thank you very much for the suggestion. The estimated attrition rate of 30% in this project is higher than we had in previous research projects with overweight populations (NEUROECOBE and BRAINOBE projects). The actual retention rate we are having so far is 21% at 3-month follow-up and 25% at 6-month follow-up. Therefore, we believe that the estimate reflected in the manuscript could be quite tight. Participants are highly motivated to participate in the intervention, and even the control group, which receives a personalised diet and physical exercise recommendations, shows good adherence to the programme.

Point 2: There should be introduced a method of evaluating the real readings of the pedometers as some passive activities might generate false movement. Where smartwatches are available, maybe it would be useful to use the body composition analysis already available on the smartwatch and also collect this data in relation to bodyweight and BMI.

Response 2: Thank you very much for the suggestion. Body composition parameters (fat percentage, water percentage and muscle mass percentage) are obtained from a pharmacy digital scale for all participants, and now this information has been included into the manuscript (page 7).

On the other hand, false movement is a limitation of pedometers, but hopefully it will affect all groups evenly. Nevertheless, we also consider other measures related to physical activity to guarantee it is correctly measured: International Questionnaire about Physical Activity (IPAQ), physical exercise habits with a visual analog scale, Physical Exercise Intentions Scale (PEIS), and adherence to physical exercise recommendations with a visual analog scale

Point 3: Some corrections of the English language should be done. For example, "pedometer" is the correct term (not podometer) that should be used in the table describing the trial interventions and proposed data collection; line 242 "0 = not healthy at all", or line 310 "How much do you want ultra-processed food?" Please thoroughly revise the questions and questionnaires that will be presented to the participants before starting the recruitment process.

Response 3: Thank you very much for your attention to the errors in the English wording. We have thoroughly revised the manuscript to correct the English wording.

Reviewer 2 Report

1.      In the section of the introduction, I suggest the author could provide further information about the theory of cognitive-behavioral therapies and help the readers’ appropriate cognition of the advantages and useful value of this issue.

2.      Also, I suggest the author provide more information about the useful effects of cognitive-behavioral therapy on obesity prevalence.

3.      What is the TRAINEP? I suggest the author could explain this keyword and provide more theoretical or practical information about this keyword.

4.      On page 4, I suggest the author could revise the figure consciously and help the readers understand the research phases. The detailed information could be rewritten and moved to the other appropriate section.

5.      In section 2.4. Outcome measures and other results, I suggest the author could provide more theoretical criteria on the cognitive-behavioral therapy and help the readers understand the useful effects or values on this strategy for obesity prevalence.

6.      I suggest the author could reorganize the related information and write a new section of discussions or conclusion. This section could help the readers understand the practical effects of TRAINEP and future suggestions for the researchers.

Author Response

Points 1 & 2:  In the section of the introduction, I suggest the author could provide further information about the theory of cognitive-behavioral therapies and help the readers’ appropriate cognition of the advantages and useful value of this issue.

Also, I suggest the author provide more information about the useful effects of cognitive-behavioral therapy on obesity prevalence.

Responses 1 & 2: Thank you very much for your suggestions. We have included more information on cognitive behavioural therapies. Extensive information on cognitive training was also included in the manuscript (pages 2-3) to help readers understand the value of these strategies and to differentiate them from cognitive-behavioural therapies.

To our knowledge, there are no data on the effects of cognitive behavioural therapy, or cognitive training, on the prevalence of obesity.

Point 3: What is the TRAINEP? I suggest the author could explain this keyword and provide more theoretical or practical information about this keyword.

Response 3: TRAINEP is the acronym for Program for the comprehensive cognitive training of excess weight in Spanish (Programa para el TRATamiento cognitivo INtegral del Exceso de Peso). This information was included into the manuscript (page 4, first paragraph).

Point 4:  On page 4, I suggest the author could revise the figure consciously and help the readers understand the research phases. The detailed information could be rewritten and moved to the other appropriate section.

Response 4: Thank you very much for the suggestion. We have removed redundant information from the figure and moved the extra information to the text, so we think it is now easier to understand.

Point 5:   In section 2.4. Outcome measures and other results, I suggest the author could provide more theoretical criteria on the cognitive-behavioral therapy and help the readers understand the useful effects or values on this strategy for obesity prevalence.

Response 5: We are not sure if we understood the comment correctly, but we believe we could have already addressed this suggestion in response to your previous comments 1 and 2. We think it solves your suggestion.

Point 6:  I suggest the author could reorganize the related information and write a new section of discussions or conclusion. This section could help the readers understand the practical effects of TRAINEP and future suggestions for the researchers.

Response 6: Thank you for this suggestion which will certainly improve the manuscript. A concluding paragraph has been included on page 13.

Reviewer 3 Report

The study protocol presented in the paper could be an interesting interventional study in a population that need more evidence of effective treatments. The paper is well written, however, there are several font and size changes. I have some methodological suggestions for the authors: 

- you exclude patients with drug treatment with an effect on eating, please be more clear and precise. Psychiatric drugs like fluoxetine could reduce the food intake, as well as olanzapine could increase it: are they excluded? 

- are the use of drugs and the psychiatric diagnosis evaluated only by self-report questionnaires? 

- BMI included is very different (from 25 - that is overweight and that could be a normal weight in men - to 40) I think this could be a problem in the analysis.

- I would use a different term that "executive function" because you used self-report questionnaires and not specific tasks and could be misleading

- I suggest to the authors to consider a correction for multiple analyses because they evaluate a large number of variables.

Author Response

Point 1: You exclude patients with drug treatment with an effect on eating, please be more clear and precise. Psychiatric drugs like fluoxetine could reduce the food intake, as well as olanzapine could increase it: are they excluded? 

Response 1: Patients with psychiatric treatment or any other medication that could affect food intake were excluded from the study. Thank you for your suggestion, we have clarified this in the manuscript (page 5, last paragraph).

Point 2: Are the use of drugs and the psychiatric diagnosis evaluated only by self-report questionnaires? 

Response 2: Drug use and psychiatric diagnosis are only assessed by self-report questionnaires. However, individualised clinical interviews may be conducted when research psychologists suspect a condition of study exclusion. This information has now been included in the manuscript (page 5, last paragraph).

Point 3:  BMI included is very different (from 25 - that is overweight and that could be a normal weight in men - to 40) I think this could be a problem in the analysis.

Response 3: We agree that differences in BMI could be a problem. Therefore, we have controlled for them in the randomisation using a minimisation process (page 5, first paragraph). Thank you for your suggestion, we have clarified this in the manuscript and possible differences in programme outcomes due to BMI will be explored (page 12, last paragraph).

Point 4: I would use a different term that "executive function" because you used self-report questionnaires and not specific tasks and could be misleading

Response 4: We agree the term could be misleading, and have changed it to Self-reported Executive Functioning to be more precise.

Point 5: I suggest to the authors to consider a correction for multiple analyses because they evaluate a large number of variables.

Response 5: Corrections will always be considered for multiple comparisons (see page 13, second paragraph).

Round 2

Reviewer 2 Report

The author mostly incorporated my suggestions and revised the manuscript. The manuscript could be published in this form for the journal.

Reviewer 3 Report

I think the authors have addressed all my concerns.